# Neural Communication Systems with Bandwidth-limited Channel

## Abstract

Reliably transmitting messages despite information loss due to a noisy channel is a core problem of information theory. One of the most important aspects of real world communication is that it may happen at varying levels of information transfer. The bandwidth-limited channel models this phenomenon. In this study we consider learning joint coding with the bandwidth-limited channel. Although, classical results suggest that it is asymptotically optimal to separate the sub-tasks of compression (source coding) and error correction (channel coding), it is well known that for finite block-length problems, and when there are restrictions to the computational complexity of coding, this optimality may not be achieved. Thus, we empirically compare the performance of joint and separate systems, and conclude that joint systems outperform their separate counterparts when coding is performed by flexible learnable function approximators such as neural networks. Specifically, we cast the joint communication problem as a variational learning problem. To facilitate this, we introduce a differentiable and computationally efficient version of the bandwidth-limited channel. We show that our design compensates for the loss of information by two mechanisms: (i) missing information is modelled by a prior model incorporated in the channel model, and (ii) sampling from the joint model is improved by auxiliary latent variables in the decoder. Experimental results justify the validity of our design decisions through improved distortion and FID scores.

## 1 Introduction

The 21st century is often referred to as the information age. Information is being created, stored and sent at rates never before seen. To cope with this deluge of information, it is vital to design optimal communication systems. Such systems solve the problem of reliably transmitting information from sender to receiver given some form of information loss due to transmission errors (i.e. through a noisy channel). As the size of the transmitted messages goes to infinity for memory-less communication channels, the joint source-channel coding theorem (Shannon, 1948) states that it is optimal to split the communication task into two sub-tasks: (i) removing redundant information from the message (source coding) and (ii) re-introducing some redundancy into the encoded message to allow for message reconstruction despite the channel information loss (channel coding). As a result, separate systems have been studied extensively in the literature and in fact are the standard way of coding for many scenarios. However, it is also well known that there are limits to the optimality of separate systems in practical settings. Most importantly for this work, limitations arise when we seek to encode finite length messages (Kostina & Verdú, 2013). These limits lead to two consequences: (i) When there is a budget on transmission bits, source and channel coding errors need to be balanced for best reconstruction results. (ii) Decoding via maximum-likelihood principle becomes an NP-hard problem (Berlekamp et al., 1978). Thus approximations need to be made that can lead to highly sub-optimal solutions (Koetter & Vontobel, 2003; Feldman et al., 2005; Vontobel & Koetter, 2007).

Recent work (Choi et al., 2019; Farsad et al., 2018), has thus looked at the problem of learning to jointly communicate. This includes systems that learn to do source and channel coding jointly from data. Practically this can be achieved by learning neural network encoders and decoders, where channels are simulated by adding noise to encoded messages. Several desirable properties of such systems were shown, including improvements in decoding speed and code length. Complementary to this body of work, we focus this study on the investigation of neural joint models with the bandwidth-limited channel. Specifically, we direct our experimentation on the bandwidth-limited channel due

to it's ubiquity as a fundamental component in the real world communication systems. The main contributions of this work include:

1. We cast the problem of learning joint communication as a variational learning problem, parallel to other work (Choi et al., 2019).
2. We justify the importance of jointly learned systems by empirically evaluating the gap between neural systems for joint and separate communication.
3. We design standard channels such as the Gaussian and Binary channel as differentiable probabilistic nodes, which serve as base for our design of the bandwidth-limited channel.
4. We investigate two core design choices of our neural joint model and bandwidth-limited channel: (i) how transmission rate can be improved through learned prior models (ii) how we may improve image reconstructions in the low bandwidth regime through the introduction of auxiliary latent variables in the decoding process.

## 2 NOTATION AND PRELIMINARIES

We mark sets as calligraphic letters (i.e. $\mathcal{X}$), random variables as capital letters (i.e. $X$) and their values as lower case letters (i.e. $x$). We use capital letters to denote probability distributions (i.e. $P(X)$) and lower case letters for the corresponding densities (i.e. $p(x)$). We will refer to a property of stochastic processes, the entropy. It describes the average rate at which a process emits information. Formally,

$$H(X) = \mathbb{E}_{P(X)}[-\log P(X)], \tag{1}$$

where $\mathbb{E}$ is the expectation. Further, we expect the reader to be familiar with the distortion-rate theory. Appendix B summarizes these shortly and makes connections to neural compression systems.

## 3 SOURCE AND CHANNEL CODING FOR COMMUNICATION SYSTEMS

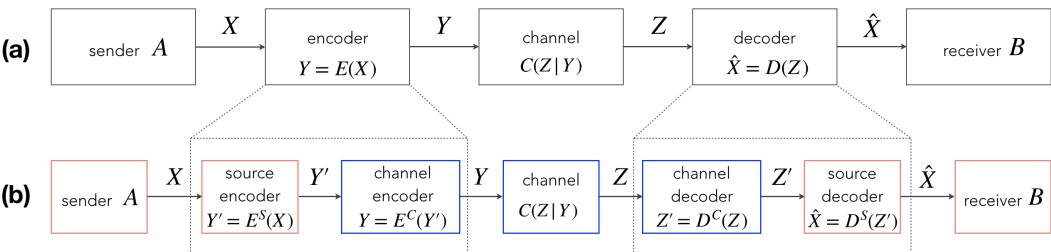

Figure 1: **(a)** Joint communication system: A message $X$ passes through a joint source and channel encoder before it passes a channel and is subsequently decoded. **(b)** Separate communication system: This system distinguishes source encoding and decoding (red) from channel encoding and decoding (blue). Red and blue system are designed independently of each other.

In this section the reader will be introduced to communication systems and in particular to the challenge of joint coding when a finite bit-length budget is given.

Communication is defined by an entity $A$ called the sender, or source, that induces a state $X$, the message, in another entity $B$, the receiver. We call this transfer of information successful if $A$ and $B$ agree about the message being sent: $X = \hat{X}$, or if the message distortion $||X - \hat{X}||$ does not exceed a certain level $D$. Real-world communication is an inherently noisy physical process where many uncontrollable or unpredictable factors may interfere with a sent message before it reaches its receiver. To account for this interference, communication is typically organized into three distinct components which we illustrate in Figure 1: (i) The *encoder* $Y = E(X)$, whose role is to compress its inputs (i.e. to remove redundant information) and subsequently prepare them for transmission through the channel with minimal distortion. (ii) The *channel* $Z = C(Z|Y)$ over which we have no control, and represents the unpredictable distortions caused by the physical transmission process. (iii) The *decoder* $\hat{X} = D(Z)$, whose goal is to reverse the process to the original datum $X$ from the received code $Z$.

**Channel capacity** The most important characteristic of a channel is its capacity. In order to evaluate it, we may compute the number of distinguishable messages that we can send through the channel given an encoder. The logarithm of that number is referred to as *information capacity of the channel*. It is given by the maximum of the mutual information of $Y$ and $Z$, $I(Y; Z)$, taken over all possible input distributions $P(Y)$,

$$C = \max_{P(Y)} I(Y; Z). \tag{2}$$

Other relevant properties of channel models include (i) *bandwidth*, the number of information units passing a channel per time unit, (ii) *memory*, the independence of joint probabilities of a transmitted sequence, where a channel with fully independent joint probabilities is called 'memoryless', and (iii) *feedback*, the ability for the sender side of the system to know what bits have arrived at the receiver side, resulting in $Y = E(X, Z)$. Note that in this work, we will constrain our research question to feedback and memoryless channels.

**Joint source-channel coding** The channel capacity fundamentally restricts the ability of a communication system to transfer messages. The source-channel coding theorem (SCCT) specifies this restriction as follows. For i.i.d. variables and memoryless channels, given a certain tolerated message distortion $D$, we must send codes with length $R(D)$. The data may be recovered by the receiver at distortion $D$ if and only if $R(D) < C$.

Furthermore, it can be shown that there exists a two stage method that is as good as any alternative to transmit information over a noisy channel reliably: source coding and channel coding. These two steps can be accomplished by two distinctly designed systems, referred to as the source encoder and decoder, $E^S(\cdot)$ and $D^S(\cdot)$, and the channel encoder and decoder, $E^C(\cdot)$ and $D^C(\cdot)$, respectively. It is easy to see why this result had great impact on the design of communication systems in practice. A communication problem is essentially defined by its source and its channel. Any such tuple defines an individual problem, resulting in an enormous problem space. By separation, it is possible to independently reuse good source or channel solutions for other problems.

However, there are also restrictions to the applicability of the SCCT. For finite length messages, we have to trade bits for compression and channel coding against each other. This is not trivial (Pilc, 1967; Csiszar, 1982; Kostina & Verdú, 2013). On top of this, encoders and decoders

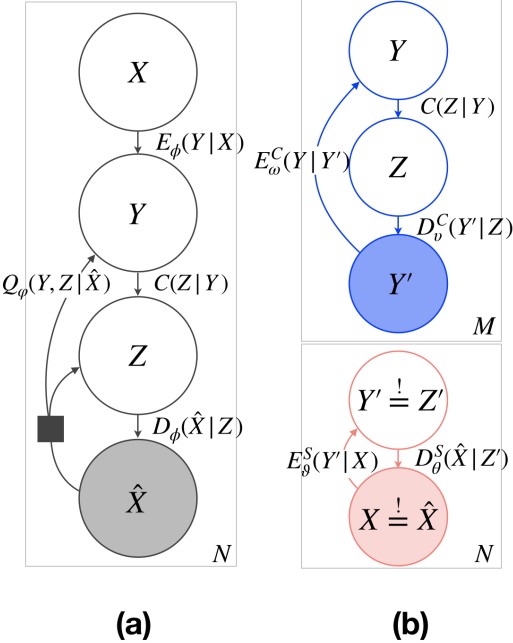

**(a)**          **(b)**

Figure 2: **(a)** *Graphical model of the jointly learned communication system.* The message $X$ is passed by the encoder and the channel, to be reconstructed into $\hat{X}$. Note that because marginalization is not possible we apply a variational approximation $Q$ to aid inference. **(b)** *Graphical models of the separately learned communication system.* Two systems are learned independently: a VAE for source compression (red) and an AE for channel transmission (blue).

are being idealized to be any function. In practical settings however, we may not be able to identify optimal encoders. Further, they are computationally restricted. In the era of machine learning, however, hypothesis spaces can be searched increasingly quickly in an automated fashion, allowing researchers to search over the space of joint solutions for the first time. For these reasons, we propose to learn joint communication systems using flexible function approximators such as deep neural networks.

## 4 LEARNED COMMUNICATION SYSTEMS

In this section, we outline how to learn neural encoders and decoders for a given joint communication problem defined by a channel and a source. Our approach requires a differentiable path through the communication system. For this, we design appropriate channel models. Additionally we introduce a new design for the bandwidth-limited channel, adapted from classical models, and explain how to do marginalization of bands in practice. Consequently, we frame learning in the joint and in the separate model as a variational optimization problem. Our approach is related to auto-encoders (Vincent et al., 2010) and variational auto-encoders (Rezende et al., 2014; Kingma & Welling, 2013). We will outline the connection here. Finally, we introduce auxiliary latent variables (ALV) to the decoding process as means to combat low reconstruction quality when little information is transmitted though such a model.

### 4.1 CHANNEL MODELS

To enable back-propagation through a communication system, we shall introduce the most common channel models in the literature and explain how to build them in a differentiable fashion.

**Gaussian Channel** We start this discussion with the Gaussian channel model, the most important continuous alphabet channel. It is a time discrete channel that distorts incoming signal Y by i.i.d. Gaussian noise W.

$$Z_i = Y_i + W_i, \qquad\qquad W_i \sim \mathcal{N}(0, \sigma^2) \qquad\qquad (3)$$

However, this particular definition is of limited use. When the noise is fixed but the power of the input is not, one can easily design channel encodings that essentially ignore that noise. It is thus common to power constrain the input, this is equivalent to keeping a constant *signal-to-noise ratio* (SNR), $s$. It can be shown that the channel capacity of a power limited Gaussian channel is equal to $C = \frac{1}{2} \log(1 + s)$ bits per transmission (Cover & Thomas, 2012). For a differentiable Gaussian channel with constant SNR, we assume the channel input to be an isotropic Gaussian distribution $Y_i \sim \mathcal{N}(\mu_{Y_i}, \sigma_{Y_i}^2)$. We propose to use the reparameterization trick (Kingma & Welling, 2013; Rezende et al., 2014), where a probabilistic node is separated into a parameter independent stochastic node and a deterministic one. By using the trick twice we can rewrite the channel to

$$Z_i = Y_i + \frac{\mu_{Y_i}}{s} \cdot W_i, \qquad\qquad W_i \sim \mathcal{N}(0, 1). \qquad\qquad (4)$$

**Bandwidth-limited channel** Related to the Gaussian channel, and one of the most important models for communication, e.g. over a radio and wifi, is the bandwidth-limited channel. The channel capacity for a Gaussian bandwidth limited channel is known to be linearly related to the bandwidth $C \sim B$. In the classical literature, this is described as a continuous time, white noise and bandwidth-pass filtered channel [1]; however, in this work, we adopt the concept to be a discrete time channel, for which we introduce the bandwidth $B$ as a discrete latent variable,

$$C(Z|Y) = \sum_B C(Z, B|Y) = \sum_B P(B) \underbrace{\prod_{t=1}^{B} C(Z_t|Y_t, \{Z_\tau\}_{\tau=1}^{t-1}) \prod_{t=B}^{T} P_{Y_t}(Z_t)}_{=C(Z|B,Y)}. \qquad (5)$$

where $t$ is a discrete time step. In words, a signal $X$ gets encoded into a sequence $Y = \{Y_t\}_{t=1}^T$. The sequence gets transmitted up to $B$ by sending $Y = \{Y_t\}_{t=1}^B$ though a channel $C(Z_t|Y_t)$ such as the Gaussian channel. Other information $Y = \{Y_t\}_{t=B+1}^T$ is lost. This information is replaced by samples from a prior over $Y_t$, $P_{Y_t}(Z_t)$. The full integration over the input domain required to compute the integral $P_{Y_t}(Y_t) = \int e(Y_t|x_t)\, dx$ is expensive. Thus, we will introduce an approximation to it, i.e., a standard Gaussian prior or a more elaborate model such as the ConvDraw prior (Gregor et al. (2016)).

To summarize, we have introduced a differentiable and computationally efficient version of the bandwidth-limited channel. For this, we turn it into a time discrete channel by introducing the discrete latent variable $B$. To marginalize over the latent variable we may either do Monte Carlo sampling or

---

[1]see Cover & Thomas (2012), chap. 10 for more details

complete marginalization. The model also requires a model for codes that have been dropped. This is similar to the prior in a variational auto-encoder and can be learned to arbitrary complexity.

Other differentiable models include the erasure channel, first considered in Kim et al. (2018a). However, this channel is mainly relevant for feedback systems, we will thus not discuss it in this context. Another relevant channel is the Binary channel, which we detail in the appendix. For real-world channels there is the option to learn a parametric model that emulates them by sending random information units. Subsequently this model can be utilized as channel model. If only a black-box model of the channel is available, our proposed framework may be extended by using discrete optimization schemes. For example VIMCO (Mnih & Rezende, 2016) has been used in Choi et al. (2019). The implementation of the channels we consider here can be found online, `github.com/anonymous_code`.

## 4.2 SEPARATE SOURCE-CHANNEL CODING

As described in section 3, the joint communication problem can be broken down into two independent problems; the source coding and channel coding problem. Here, we demonstrate how to apply the variational auto-encoder as a source coder and an auto-encoder as channel coder. Note that, there is no exchange of information between those two systems. We provide a visual aid for this section in Figure 2.

**Source-VAE** In recent years, neural networks have been shown to be useful source compressors. Specifically, variational auto-encoders (VAEs) have been pointed to as natural source coding systems (Kingma & Welling, 2013; Alemi et al., 2017), showing great practical success (Ballé et al., 2016; 2018b; Minnen et al., 2018; Zhou et al., 2018; Tschannen et al., 2018). Such a source-VAE is essentially a learned probabilistic model. Based on a set of samples emitted by the source $\mathbf{X} = \{x_n\}_{n=1}^N$, we aim to learn the source encoder $E_\vartheta^S(Y'|X)$ and the source decoder $D_\theta^S(\hat{X}|Z')$, both parameterized functions of $\vartheta$ and $\theta$, respectively. The learning objective thereby originates from looking at the model as a latent-variable model (with the encoding $Z$ being the latent variable) for which we aim to do maximum marginal likelihood learning of the parameters. The involved marginalization, however, forces the introduction of a variational approximation, the encoder, to construct a lower bound on the marginal log likelihood, known as variational inference. For this, we set the source encoder to be the variational approximation to the source decoder, such that $Y' \stackrel{!}{=} Z'$ and $X \stackrel{!}{=} \hat{X}$:

$$\mathbb{E}_{P(X)}\left[\log P(X|\theta,\vartheta)\right] \geq \mathbb{E}_{P(X)}\left[\underbrace{\mathbb{E}_{E_\vartheta^S(Y'|X)}[\log D_\theta^S(\hat{X}|Z')]}_{=:-D} - \underbrace{KL(E_\vartheta^S(Y'|X)||P_\theta(Z))}_{=:-R}\right] \quad (6)$$

This bound is known as the evidence lower bound (ELBO). Optimizing ELBO is equivalent to optimizing a rate($R$)-distortion($D$) problem. We can adjust the rate-distortion trade-off to a desired rate or distortion by introducing a parameter $\beta$ into the objective, this framework is well known as $\beta$-VAE (Higgins et al., 2017; Alemi et al., 2017).

Generally, it is possible to optimize decoder and encoder independently. This however would only make sense if we consider channel coding systems that do not try to reconstruct their inputs. Note that in contrast to the original formulation in section 3, encoder and decoder have been turned into probabilistic mappings rather than deterministic ones. This allows one to find an ideal compression rate given a certain distortion-rate trade-off $\beta$. The rate can practically be achieved with the so called bits-back coding (Hinton & Van Camp, 1993; Townsend et al., 2019). For inference it became common that the parameters for the encoder distribution may be predicted by a neural network parameterized by $\vartheta$. This is called amortized inference. The parameters of this inference model and the generative model, the decoder, are trained jointly though stochastic maximization of ELBO. To do this efficiently, it is common to use the reparameterization trick (Kingma & Welling, 2013; Rezende et al., 2014).

Finally, it is important to note that the prior distributions in the context of compression may not be learned $P_\theta(Z) = P(Z)$ . This would conflict the independence of the source and channel.

**Channel-AE**   For training a neural channel coding system, we will use samples from the source independent prior $\{y'_m\}_{m=1}^M$, $y'_m \sim P(Z)$. After using a deterministic encoding $Y = E_\omega^C(Y|Y')$, we send $Y$ though the probabilistic channel $Z \sim C(Z|Y)$, after which we try to recover the inputs by channel decoding $Z' = D_\upsilon^C(Z'|Z)$. The system is trained by minimizing a measure of distortion between $Y'$ and $Z'$.

Note, that a for a simple additive withe Gaussian noise channel there exists a near optimal channel coding scheme: LDPC. However, in more general scenarios they do not perform as well anymore and can be beaten by neural network architectures (Kim et al., 2018a). Further, it has been shown that neural networks can decode them efficiently (Nachmani et al., 2016). For the sake of generalizing to more complex channels we thus propose general purpose neural network channel coding.

### 4.3   JOINT SOURCE-CHANNEL CODING

For the jointly optimized system, we translate the communication system as described in section 3 into a generative model $P_\phi(\hat{X}|X) = \int e_\phi(y|X)c(z|y)d_\phi(\hat{X}|x)\,dy\,dz$. Similar to the previous section, we think of the encoder and decoder as parameterized mappings, while the channel model is taken as given. We are interested in performing maximum likelihood learning of the model parameters $\phi$, by optimizing

$$\mathbb{E}_{P(X)}\left[\log P_\phi(\hat{X}|X)\right] = \mathbb{E}_{P(X)}\left[\log \int e_\phi(y|X)c(z|y)d_\phi(\hat{X}|x)\,dy\,dz\right] \tag{7}$$

The required marginalization in equation 7, however, leads to generally intractable integrals. One frequently applied solution is to introduce a variational approximation $Q_\varphi(Y, Z|\hat{X})$ to the posterior, to construct a lower bound on the marginal likelihood.

$$\log P_\phi(\hat{X}|X) \geq \mathbb{E}_{Q_\varphi(Y,Z|\hat{X})}[\log D_\phi(\hat{X}|Z)] - D_{\mathrm{KL}}(Q_\varphi(Y,Z|\hat{X}) \,\|\, E_\phi(Y|X)C(Z|Y)) \tag{8}$$

As before, this represents an ELBO. Note though that, the first term in equation 8 refers to the quality of the message reconstruction and the second to how closely the receiver understands the sender. This is different to the previous section where the message never actually passes the communication system. The variational posterior plays a very different role there where it is assumed to be the encoder. In the joint scenario the posterior only serves to train the system, at test time, however it is of no interest. To sum it up, our proposed framework optimizes the actual objective of communication, the message reconstruction. For channels that do not allow for information transfer this model turns into a VAE.

**Auxiliary latent variable Decoders**   When the information transmitted by the channel is variable, i.e. for the bandwidth-limited channel, a model has to adapt to low and high information transmission rates. To contest information loss due to a noisy channel, we propose to introduce auxiliary latent variables $V$ to the decoder model. This model choice acknowledges the implicit marginalization over lost information. Although expected message distortion is unchanged, when sampling from such a model, message reconstructions should occur more in distribution with the true source distribution (e.g. one would expect sharper images).

We can enforce this change to the decoder by adapting the distortion term in equation 8. As before we would need to marginalize over $V$ but choose the variational approach instead,

$$-D \geq \mathbb{E}_{Q_\varphi(Y,Z|\hat{X})}[\mathbb{E}_{Q_\xi(V|\hat{X},Y,Z)}[D_\phi(\hat{X}|Z,V)] \tag{9}$$
$$- D_{\mathrm{KL}}(Q_\xi(V|\hat{X},Y,Z)|P(V))].$$

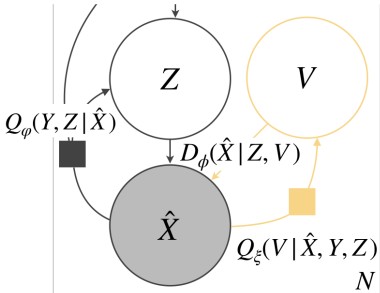

Figure 3: Excerpt of the graphical model in Figure 2 **(a)**. We show how the decoder changes when introducing auxiliary latent variables $V$.

Again, we introduced an approximate posterior to circumvent the intractable task, where $P(V)$ is a prior over these newly introduced latent variables. Just as before, the parameters of the variational

distribution shall be inferred by a deep neural network with parameters $\xi$. We indicated the components introduced to the communication model in yellow in Figure 3. We note that we could execute the same idea using other conditional generative models, and corresponding inference methods, such as conditional GANs. We leave this exploration to future research.

## 5 RELATED WORK

The field of learned image and video compression has enhanced rapidly over the past few years. While Ma et al. (2019) give a recent concise overview of the field, here we focus on probabilistic auto-encoding approaches first proposed by Theis et al. (2017). The main focus of the field of image compression is to close the gap between theoretical ideas and well performing systems. One block of efforts focuses on learning representations. While VAEs tend to work better in the continuous regime, most codes and channels can best be described by binary representations. To bridge this, it has been proposed to (i) quantize continuous representations by convolving them with a uniform distribution (Ballé et al., 2016; 2018b; Minnen et al., 2018; Agustsson et al., 2017), (ii) learn discrete representations directly (van den Oord et al., 2017; Ballé et al., 2018a; Shen et al., 2019) or even (iii) learn to generate common codecs e.g. JPEG (Jiang et al., 2017; Liu et al., 2018). Note that some of these systems rely on learned priors; however, these are actually not suitable for separate coding. Other work is focused on biasing compression towards image features important for perception or system security (Li et al., 2018; Agustsson et al., 2018). For situations where sequences of source inputs are communicated, neural buffers have also been explored to allow reordering of elements to improve code length (Graves et al., 2018). Another branch of research focuses on the architecture of encoder and decoder models (Gregor et al., 2016; De Fauw et al., 2019; Zhou et al., 2018). Additionally, there is work looking into performing tasks on compressed representations directly (Torfason et al., 2018). Important to mention also are efforts to make the often expensive encoder and decoder more computationally efficient (Ballé et al., 2015).

In contrast to neural source coding, neural channel coding has yet to be explored so extensively. However, first studies (Nachmani et al., 2016; Gruber et al., 2017; Cammerer et al., 2017; Dörner et al., 2017) demonstrate great success with neural encoder/decoder architectures. For example, it was shown that a neural model can find a solution to the Gaussian feedback channel which benefits from the feedback, a result known before but not demonstrated by any channel code yet (Kim et al., 2018a;b).

Most related to our work in spirit is a range of end-to-end learned joint communication systems. Farsad et al. (2018) apply a joint source channel system to text; Bourtsoulatze et al. (2019) use auto-encoders to transmit messages over the AWGN channel; and Zarcone et al. (2018) use joint systems for data compression. Closest to our work is the study by Choi et al. (2019) where they learn the communication system in a variational fashion as well, but exclusively look at the binary erasure channel. The discrete channel leads to another variant of the learning scheme.

## 6 EXPERIMENTS

We focus our experiments on the bandwidth-limited channel with additive white Gaussian noise (AWGN) and power restricted inputs. The latter ensures a limited channel capacity. First, we verify the importance of joint coding in contrast to separate channel coding in this context. For this, we compare neural joint and separate models, finding the joint model consistently outperforms it's separate counterpart. These findings echo other recent work. We therefore expand upon this by focusing the remainder of our experiments on the bandwidth-limited channel model proposed in this work. Specifically, we investigate the performance of a neural joint model with the AWGN bandwidth-limited channel for different prior and decoder choices.

All results are evaluated on CelebA (Liu et al., 2015). All images were re-scaled to a resolution of $32 \times 32$ pixels. Encoders and decoders have generally been chosen to be Residual Networks (He et al., 2016), due to their wide usage in a range of generative modelling tasks, e.g. in Gregor et al. (2016).

### 6.1 COMPARING JOINT AND SEPARATE NEURAL MODELS WITH GAUSSIAN CHANNEL

As previously discussed in section 3, we can not predict precisely how a separate model would compare in contrast to a joint one. We hence compare separate and joint neural models as described in section 4 for the AWGN at various SNRs. For both models we choose the same posterior distribution for latent encoding: isotropic Gaussians. We additionally choose the observation model to be Gaussian since it is quite common to measure distortion in L2-space. Encoder and decoders of both models share the same architecture configuration. For the separate system, we choose a standard Gaussian to be the prior for the source-VAE and simultaneously the data source for the channel-AE. We note that this cannot be a data dependent prior as this would leak information to the channel coding system. For both models we hyper-optimize over a range of beta values on a log-scaled grid[2]. We optimize both models with an SGD algorithm.

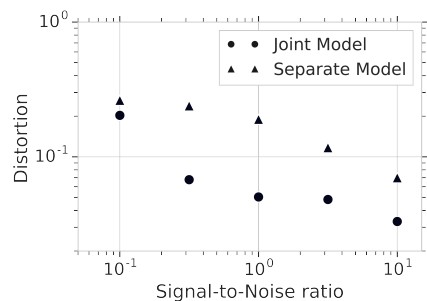

Figure 4: Results of comparing distortion for joint and separate neural communication systems at various signal-to-noise ratios for the Gaussian channel. The joint model outperforms the separate one consistently.

We evaluate both systems by sending a message through the encoder, channel and eventually the decoder, subsequently measuring the L2-distortion between sent and received messages. The quantitative results are presented in Figure 4 on the left. For any of the 5 SNRs that we run our experiment on, we find the joint model outperforms the separate one. We observe, though, that the difference between the systems shrinks towards either end of the range of the SNRs presented. This effect can be explained: For very high SNRs the channel model becomes somewhat redundant, thus both systems resemble a source-VAE. At very low SNRs both systems fail to communicate as they approach the channel capacity. Our findings are in line with other recent work: Choi et al. (2019) show that joint systems outperform hybrid models (neural source coding, hand-designed channel coding) and Kim et al. (2018a) show, for some feedback channels, learned neural models outperform hand-crafted channel codes.

## 6.2 COMMUNICATION MODEL DESIGN FOR BANDWIDTH-LIMITED CHANNEL

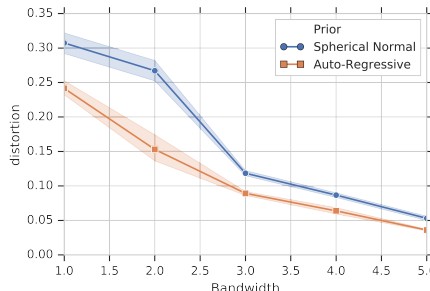
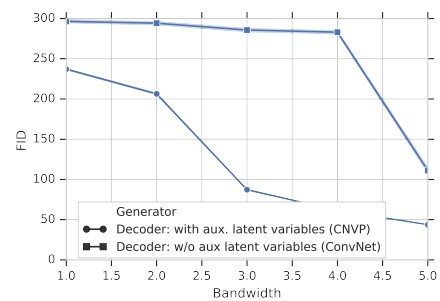

Figure 5: We consider joint models trained based on the AWGN bandwidth-limited channel with a fixed SNR of 1. In both figures we contrast message quality with bandwidth. The higher the bandwidth the more information is transmitted to the receiver. *Left:* We measure message quality by distortion in L2-space. We compare two approximations to the channel encoding distributions. Our complex prior outperforms a simpler one. Further, we observe a linear relationship between bandwidth and distortion. *Right:* We measure message quality by FID score. Lower FID score is better. We compare decoders without auxiliary latent variables to decoders with auxiliary latent variables.

After verifying the importance of joint modelling for Gaussian channels, we will now investigate the performance of a joint model on the AWGN bandwidth-limited channel design we introduced in section 4.1. In this experiment we fix the SNR of the AWGN to 1.

---

[2]To get an understanding of the sensitivity to this parameter we put all results in the appendix.

Two choices for the model are relevant, the prior that models channel codes and the decoder. Both deal with a lack of information in the low bandwidth regime.

**Prior** As mentioned in the section 4.1, we require an approximation to $P_{Y_i}$. In our first experiment, we investigate how much the complexity of this approximation influences the quality of message reconstruction. Here we shall compare a spherical Gaussian and ConvDraw prior (see Gregor et al. (2016)) to contrast a simple with a complex approximation. We consider a 100 dimensional latent space. The space is partitioned into 5 parts. Each part representing another band. Other specifications of the experiment are equivalent to the previous section. We present our findings in Figure 5 (Left). We observe that, as expected, the message distortion decreases when we transmit more information, for both approximations. We additionally observe that the quality of reconstruction increases when the more complex prior model is used, and the distortion gap between priors increases when less information units are being transported through the channel. Furthermore, for both prior choices, the distortion decreases almost linearly with the bandwidth increasing. This result is in line with classic findings that show a linear relationship between channel capacity and bandwidth of an input power restricted AWGN. Finally, we shall give a visual impression of the reconstructions at various bandwidth in Figure 6 in the appendix.

**Decoder** For small bandwidths, we find that loss of information leads to blurry reconstructions even with learned priors. To combat this, we contrast a model without auxiliary latent variables with our proposed auxiliary latent variable model. Specifically, for these two models, we use an unconditional ConvDraw decoder and a conditional ConvDraw decoder (Gregor et al., 2016) respectively. As a measure of in-distribution affiliation we use the well established FID measure (Radford et al., 2015). This measure has mainly served to evaluate the quality of GAN samples. Smaller FID measures are better. In this experiment, we use the more complex auto-regressive prior model. Other experiment details remain the same as before. The results of this experiment are presented in Figure 5 (Right). For both decoders, as expected, the sample quality drops for smaller bandwidth. However, the model with auxiliary latent variables significantly outperforms the one without across the full range of bandwidth presented here. We thus conclude auxiliary latent variable decoders can significantly improve the quality of communicated messages in some respects, and therefore encourage their continued exploration.

## 7 DISCUSSION

In this paper, we derived a generative model for joint coding with the bandwidth-limited channel and showed how to perform learning based on variational inference. For this, we introduced a differentiable and efficient model of the channel. Since back-propagation through the channel is now possible, we demonstrate how we can learn flexible function approximators for coding by Monte Carlo sampling.

To justify the usage of joint coding instead of channel coding, we first compared joint with separate communication models. Joint models were shown to consistently and significantly outperform their separate counterparts. Given joint coding as a basis, we investigate our main hypothesis that when a channel transfers little or variable amounts of information, the decoder might be helped by understanding the source distribution. We put this idea into practice by focusing on two modelling choices. First, when there is no information transferred, the decoder may draw a sample from the encoding distribution $P_Y(Z_i)$ to get a source-typical encoding. We test how the complexity of the distribution model influences reconstruction performance. We find the more complex model to improve the distortion especially in the low transmission regime. Second, when sampling message reconstructions from the communication system, missing information leads to averaged reconstructions (i.e. blurry images). We prevent this by introducing auxiliary latent variable decoders. In experiments, we show that these decoders improve message reconstruction considerably in terms FID score.

Further, this models serves as a simple method to learn a latent encoding that is sorted according to information content and channel noise, eliminating the need to pass the latent code through a lossless compressor before transmitting the data. This is an essential property for sequential information transfer. In future work, we want to explore this aspect more extensively. Future efforts in this field would focus on reinforcing our finding further by investigating the same hypothesis in other data domains and with other channels.

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

# Appendices

## A    MESSAGE RECONSTRUCTIONS FOR THE BANDWIDTH-LIMITED CHANNEL

$X$

$\hat{X}(\{\})$

$\hat{X}(\{Y_1\})$

$\hat{X}(\{Y_t\}_{t=1}^2)$

$\hat{X}(\{Y_t\}_{t=1}^3)$

$\hat{X}(\{Y_t\}_{t=1}^4)$

$\hat{X}(\{Y_t\}_{t=1}^5)$

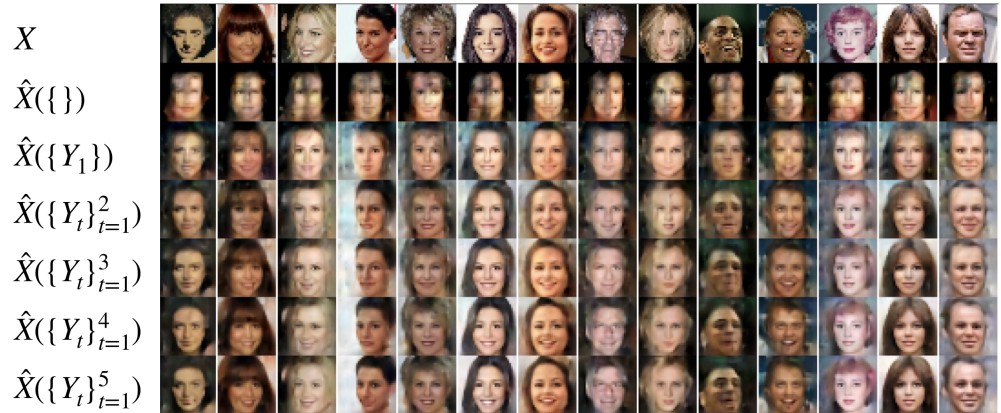

Figure 6: *First row:* Message samples from the source distribution. *Other rows from top to bottom:* Samples of the reconstructed message at all considered bandwidths. The top row has least information.

## B    RATE-DISTORTION PERSPECTIVE

Originally, information theory would study how a message can be communicated over a noisy channel to a receiver without errors. It is often a more realistic scenario, though, to think of the receiver to tolerate a certain amount of distortion. Intuitively, the more we allow for distortion of a message the smaller the number of bits we need to communicate. Rate-distortion theory is a major field of information theory that studies how these modifications to the original set-up effect fundamental theorems such as data compression or transmission. The limitations of the classical view become clear when considering continuous random variables. Continuous random variables require infinite precision to represent exactly. Hence it is not possible to send finite rate codes. Assume $X$ to be the continuous random variable to be represented by $X'(X)$. Say we are given $R$ bits to send $X$. $X'(X)$ than can take $2^R$ values. The goal of rate-distortion coding is to distribute these $2^R$ codepoints such that a minimal distortion, measured by a distortion function $d$,

$$d : \mathcal{X} \times \mathcal{X} \to \mathbb{R}^+, \tag{10}$$

where $\mathcal{X}$ is the source alphabet, is being achieved.

### B.1    SOURCE CODING IN A RATE-DISTORTION SENSE

Source coding in the context of rate-distortion theory entails two steps: quantization $X'(X)$ and traditional source encoding $Y(X') = Y(X)$. In both steps the goal is to keep the loss of information minimal given a rate that shall be achieved, $I(X;Y) \leq R$ where

$$I(X;Y) = \mathbb{E}_{P(X,Y)}[\log P(X,Y) - \log P(X)P(Y)], \tag{11}$$

is the mutual information. The goal is to keep this bound tight. However, computing the mutual information is hard since we do not have access to the true data density. Following Alemi et al. (2017), we instead find a variational approximation,

$$H - D \leq I(X;Y) \leq R \tag{12}$$

with

$$D := \mathbb{E}_{P(X)}[\mathbb{E}_{E^S(Y'|X)}[\log D^S(\hat{X}|Y')]] \tag{13}$$

$$R := \mathbb{E}_{P(X)}[\mathbb{E}_{E^S(Y'|X)}[\log E^S(Y'|X) - \log M(Y)]], \tag{14}$$

where they introduce $M(Y)$ as the variational approximation to $P(Y) = \mathbb{E}_{P(X)}[E^S(Y'|X)]$. This reestablishes that the data entropy bounds feasible (compression rate $R$, distortion $D$) pairs: $H(X) \leq R + D$. This ties together with the result from optimal coding that the source entropy bounds the optimal code length. Hinton & Van Camp (1993) show that via bits-back coding this code length (rate) can actually be achieved. This argument has further been hardened by Townsend et al. (2019) who design an actual compression algorithm in this manner.

## B.2 JOINT SOURCE-CHANNEL CODING IN A RATE-DISTORTION SENSE

In the previous section we discussed how relaxing the requirement to sending a message exactly, to sending a message under a certain distortion, effects source coding. The optimal code length would thus be $R(D)$ bits/symbol rather than $H(X) \geq R(D)$ in the error-free scenario. We can connect this information to Shannon's channel coding theorem. We know that the channel capacity $C$ restricts the number of bits that can be send. Thus there exists a solution for a maximum distortion communication system only if $R(D) < C$. When in the previous section $R$ helps to describe the number of bits that represent a random variable $X$, we can similarly find a variational approximation to $I(X, Z)$ the amount of bits representing $X$ after passing the channel,

$$T := \mathbb{E}_{P(X)E(Y|X)} \left[ \mathbb{E}_{C(Z|Y)}[\log C(Z|Y) - \log N(Z)] \right], \tag{15}$$

where equivalent to the discussion in the previous section $N(Z)$ is the variational approximation to $P(Z) = \mathbb{E}_{P(X)E(Y|X)}[C(Z|Y)]$. We shall refer to $T$ as the transmission rate.

# C RELAXING THE BINARY CHANNEL

## C.1 RELAXING THE BERNOULLI: BINARY CONCRETE DISTRIBUTION

Learning of systems with stochastic nodes $P_\theta(X)$ in Machine Learning is often synonymous with optimizing an objective function $\mathcal{L}(\theta, \phi) = E_{X \sim P_\theta(X)}[f_\theta(X)]$ w.r.t the parameters $\theta, \phi$ via some gradient descent based scheme. The challenge lies in computing the parameters $\theta$ that belong to the stochastic node. A popular approach to this problem is the application of the so called reparameterization trick (Kingma & Welling, 2013; Rezende et al., 2014) in which a stochastic node $P_\theta(X)$ with parameter dependency $\theta$ is turned into a stochastic node $Q(Z)$ without parameter dependentcy and a determined function $g_\theta(\cdot)$.

$$\mathcal{L}(\theta, \phi) = E_{X \sim P_\theta(X)}[f_\theta(X)] = E_{Z \sim Q(Z)}[f_\theta(g_\theta(Z))] \tag{16}$$

with $x = g_\theta(x)$. The remodelled stochastic node now allows for gradient based stochastic optimization via Monte Carlo sampling[3],

$$\nabla_\theta \mathcal{L}(\theta, \phi) = E_{Z \sim Q(Z)}[f'_\theta(g_\theta(Z)) \nabla_\theta g_\theta(Z)]. \tag{17}$$

For example, consider sampling from the Gaussian distribution, $X \sim \mathcal{N}(X|\mu, \sigma)$ can be replaced by sampling from a standard Gaussian $Z \sim \mathcal{N}(Z|0, 1)$ and applying $x = g_{\{\mu, \sigma\}}(z) = \mu + \sigma z$. Reparameterizing a discrete distribution such as the Bernoulli $\mathcal{B}(p)$ is not as straight forward. Maddison et al. (2016) propose reparameterization with Gumbel-Max trick. Specifically, the reparameterization is based on a logistic random variable $L \sim Logistic(L)$ as parameter-free stochastic node, the relaxed Bernoulli sample can than be attained by

$$Y = (L + \log(\alpha)/T) \tag{18}$$
$$X = \sigma(Y)$$

where $\alpha$ corresponds to the location parameter $p$ in the Bernoulli distribution, $T$ the temperature adjusts the amount of relaxiation and $\sigma$ is the sigmoid function. The density corresponding to this sampling procedure is given by,

$$p_{\alpha, T}(x) = \frac{T\alpha x^{-T-1}(1-x)^{-T-1}}{(\alpha x^{-T}(1-x)^{-T})^2}. \tag{19}$$

We shall call this the Binary Concrete distribution or relaxed Bernoulli distribution $\mathcal{B}_T(\alpha)$. It has several desirable properties.

---

[3]If there is no possibility of reparametization, one can retain to the score-function estimator, also known as REINFORCE or likelihood-ratio estimator, which allows to compute the gradient via Monte Carlo sampling. This however leads to higher variance gradients.

1. $P(X > 0.5) = \frac{\alpha}{1+\alpha}$
2. $P(\lim_{T \to 0} X = 1) = \frac{\alpha}{1+\alpha}$
3. If $T \le 1$ than $p_{\alpha,T}(x)$ is log-convex in $x$.

One problem, however, we may often be faced with is computing the log-likelihood of such a stochastic node. For example when computing the KL-divergence of a variational auto-encoder. Due to the saturation of the sigmoid function computing the log-likelihood empirically may lead to underflow issues. This is why it has been proposed to compute the log-likelihood based on the samples $Y$ before applying the sigmoid function since this is an inevitable function. The corresponding log-likelihood is given by,

$$\log g_{\alpha,T}(y) = \log T - Ty + \log \alpha - 2\log(1 + \exp(-Ty + \log \alpha)). \tag{20}$$

As an alternative we may clip the log-likelihood. This variant is easier to apply when there is no direct access to the stochastic node, we shall see what this means precisely in the next section.

## C.2 BINARY CHANNEL

The symmetric binary channel is a discrete channel with an input and output alphabet of size 2, $\mathcal{Y} \in \{0,1\}^D$, $\mathcal{Z} \in \{0,1\}^D$. The channel can be realized with Bernoulli noise on each input pixel,

$$Z_i = \frac{(2W_i - 1)}{2 \cdot (2Y_i - 1)} + \frac{1}{2} \qquad\qquad W_i \sim P_W(W_i) = \mathcal{B}(W_i|p) \tag{21}$$

where $\mathcal{B}(p)$ is a Bernoulli with $p$ the likelihood of keeping an input bit. The channel is called a symmetric channel because the probability of changing a bit does not depend on its state.

Following, we relax the channel as defined above to allow for training of differntiable communication models. For this we will utilize the relaxed Bernoulli distribution as described in the previous section. As for the Gaussian Channel we assume that that $Y_i$ can be constructed from a learned Bernoulli itself with $Y_i \sim \mathcal{B}_{T_{Y_i}}(Y_i|\alpha_{Y_i})$. In order to compute equation 8, we need to evaluate the channel density given its input $C(Z|Y)$. Since $Z$ depends deterministically on $W_i$ and $Y_i$, the channel density equals the noise density with transformed input argument $C(Z|Y) = P_W(((2Z-1)(2Y-1))/2+1)$. For Bernoulli noise, such as in the original channel formulation, the system could thus not learn. We thus propose to also adapt the noise to be a relaxed Bernoulli with probability density as in Eq. (19).

Finally, we may restrict our channel to a specific SNR. This can be computed to be

$$s = \frac{2pp_{Y_i} + 0.5 - p - p_{Y_i}}{-2p_{Y_i}p - 0.5 - p - p_{Y_i}}. \tag{22}$$

To train neural models, we will use the relaxed Bernoulli for both $Y$ and $W$, however the SNR is computed assuming both as Bernoulli distributions. This assumption is only correct for $T \to 0$. Initial experimental results, with the relaxed binary channel have been showing that optimization with this parameterisation is somewhat challenging. Our finding supports a finding in Choi et al. (2019) that focus on VIMCO instead of the re-parameterisation trick.

## D SENSITIVITY OF THE COMMUNICATION SYSTEMS TO THE HYPER-PARAMETER $\beta$

In optimizing communication systems, $\beta$ is perhaps the most important hyper parameter. This is why we present the complete set of results for experiment one in Figure 7. For the source VAE $\beta$ trades compression rate vs distortion. At maximum compression, the channel source distribution would be emulated perfectly and thus the channel AE input distribution. However, this scenario would also eliminate the mutual information between $X$ and $Y$. Thus a balance must be found by tuning $\beta$.

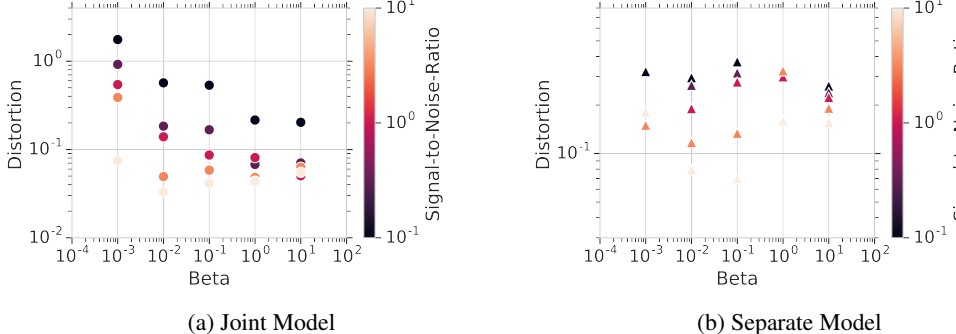

(a) Joint Model

(b) Separate Model

Figure 7: The results in Figure **??** show the distortion vs the SNR for an optimized $\beta$. Here we present all results. Note that, the joint model is more sensitive to changes in $\beta$ in the range we have chosen.

