# OpenReview forum: "Neural Communication Systems with Bandwidth-limited Channel"
_ICLR.cc/2020/Conference — Reject_

### Official Review · AnonReviewer3 · 2019-10-22
**Official Blind Review #3**

**Rating:** 6

**Review:**

This paper focuses on transmitting messages reliably by learning joint coding with the bandwidth-limited channel. The authors justify joint systems outperform their separate counterparts when coding is performed by flexible learnable function approximators. Their experiments show the advantage of their design decisions via improved distoration and FID scores.

Pros:

1. This paper is clearly written and well-structured in logic. For example, the authors use Figures 1 and 2 assist readers to catch the difference between joint communication system and separate communication system.

2. This paper gives a reliazation of joint source-channel coding, especially to give auxilary latent variable decoders.

3. This paper has been verified in both Gaussian channel and bandwidth-limited channel. The empirical results show the advantage of joint coding.

Cons:

1. Intuitively, you Section 4.3 should be better than Section 4.2. However, I don't see any difference or major items to justify this kind of benefits. Could you please explain why techniques in Section 4.3 can outperform these in Section 4.2.

2. Although the authors verified their work on CelebA, it seems that the proposed method has very limited applications. If possible, the authors should do more datasets to verify their proposed method, which will be more useful to boarder readers.

**Experience Assessment:**

I do not know much about this area.

**Review Assessment: Checking Correctness Of Derivations And Theory:**

I assessed the sensibility of the derivations and theory.

**Review Assessment: Checking Correctness Of Experiments:**

I carefully checked the experiments.

**Review Assessment: Thoroughness In Paper Reading:**

I read the paper at least twice and used my best judgement in assessing the paper.

---

> ### Author Response · Authors · 2019-11-14
> **Personal response  AnonReviewer3**
>
> Thank you for your review. In the following, we would like to address both of your questions.
>
> Question 1: Why does joint coding outperform separate coding specifically in the ML setting?
>
> This is indeed very interesting and has sparked various discussions among ourselves as well.
> There is of course classical research that illuminates why solving the joint problem may be possible in some scenarios where solving the communication problem separately is not possible (under more realistic assumptions).
>
> More specifically, in the ML context when separate coding is performed, it is understood that the channel coder receives the distribution of source embeddings. However, to be source agnostic, this distribution needs to be a generic (i.e. source data independent) distribution. For example, this could be a standard Gaussian as is used in a basic VAE. This, however, induces a bottleneck: The source coder needs to match the aforementioned generic prior distribution such that the channel coder receives the correct input. At the same time, if the source coder was to perfectly match this prior, there would be no mutual information I(source data; source embedding), and thus nothing would be learned. Joint coding does not suffer from this trade-off problem. We believe this is why it outperforms separate coding in our experiments.
>
> Questions 2a:  Relevance of the bandwidth-limited channel.
>
> We hope in our main rebuttal we could point out why modelling with the bandwidth limited channel has such a central role in modelling communication, and why introducing learning to coding is a relevant contribution.
>
> Question 2b: How does our approach extend to other domains (non-vision)?
>
> Concerning the dataset we used, we are currently running an experiment on imagenet to diversify our claim.
> We do believe, however, that our findings extend far beyond image datasets to video, language, audio and even beyond perceptual tasks. We believe that there is evidence that the architecture of the neural encoders and decoders that are employed will determine the success in these domain. Domain specific architectures are, however, not the focus of this work. Farsad et al. (2018) for example considers a language application.
>
>
> We thank the reviewer for any further feedback and are happy to discuss more if desired.

---

### Official Review · AnonReviewer2 · 2019-11-04
**Official Blind Review #1**

**Rating:** 6

**Review:**

This paper is out of my research area. I could understand that the paper studies the message transformation with bandwidth-limited channels. It seems naturally the message transformation could be represented as a autoencoder model. The paper proposed variational model for this problem and it seems to me the paper employs the popular models in neural networks for example VAE, etc. Technically, what's new of this paper? Was it the auxiliary variable decoders? Is it that this class of algorithms/models firstly applied to this problem domain? To be honest the paper mentioned most of the terminologies in ML and seems that the paper wanted to connect to them, for example, ELBO, VAE, GAN, re-parameterization, etc. The paper provides experimental results on the designed model for bandwidth-limited channel.

**Experience Assessment:**

I do not know much about this area.

**Review Assessment: Checking Correctness Of Derivations And Theory:**

I assessed the sensibility of the derivations and theory.

**Review Assessment: Checking Correctness Of Experiments:**

I did not assess the experiments.

**Review Assessment: Thoroughness In Paper Reading:**

I made a quick assessment of this paper.

---

> ### Author Response · Authors · 2019-11-14
> **Personal response to AnonReviewer2**
>
> Thank you for starting this discussion, I hope we could clarify the main points of novelty in the paper and draw the connection to existing work sufficiently in our main rebuttal (1).

---

### Official Review · AnonReviewer4 · 2019-11-21
**Official Blind Review #4**

**Rating:** 3

**Review:**


Paper Summary:

The paper proposes to use ML methods, specifically neural networks, to learn source and/or channel coding systems, either jointly or separately. Specifically, they investigate these systems under the bandwidth-limited channel.  They investigate their models applied to the task of transferring images across the channel.

Pros:

- The paper is in a difficult area, and needs to communicate ideas about communications and information as well as various deep learning models. The initial exposition does this well.

- The paper discusses ML-based communications models applied to the bandwidth limited channel and performs experiments to investigate joint vs. separate coding.

Two main things of concern:

- This area (communications, compression, coding) is highly developed, yet there are no comparisons to practical, state of the art techniques in this paper. The experiments investigate using the authors' models to send images across a noisy channel. So shouldn't we see a comparison to an equivalent non-ML-based system that currently accomplishes this task, if only for perspective?

- The paper contains too much exposition and the structure makes it difficult to read.  The authors' work seems to mostly be focused in sections 4 and 6, but prior work is summarized in section 5.  The authors want to communicate a lot of ideas which the audience might not be familiar with, and this is a difficult task. However, the paper could be reorganized such that the ideas are clearer. Rate distortion is delegated to an appendix, but two pages are spent on basic communications. Overall it could be much more focused.

Some smaller concerns:

- Pg. 3 "hypothesis spaces can be searched increasingly quickly in an automated fashion, allowing researchers to search..."
I think this statement may be over-reaching. The authors also frequently use the term "flexible function approximation". Neural networks do have interesting approximation properties. But this is not a complete picture of deep learning, and the situation is much more nuanced than this. Where is the role of optimization and data? In order to transmit data with the author's algorithms, do we need to go out and collect a large body of examples in the specific domain, like CelebA? Because you don't need to do that to e.g. compress any image with JPEG, code the data using LDPC to send across a channel. Is this comparing apples and oranges?

- Pg. 6 "Note, that a simple additive white Gaussian noise.... LDPC. However, in more general scenarios they do not perform as well and can be beaten by neural network architectures....."

I think the claims in this paragraph need to be toned down a bit. LDPC does not perform well in regards to what? Can be beaten under what conditions? Decode efficiently with regard to what block length? I don't think the picture is as clear as painted here.

- Pg. 4 The statements here regarding the bandwidth-limited channel appear to be the focus of the author's work. This section should be expanded and explained. Reading the first paragraph then going to the second paragraph ("To summarize..."), there's sort of a disconnect. How do we know these two things are equivalent? What exactly is novel that was introduced?

Small typos:
- Pg. 2 final paragraph, some  norm is used for the distortion, but this is not defined (either on pg. 2 or in App. B)
- Pg. 6 white is misspelled "withe"


**Experience Assessment:**

I have read many papers in this area.

**Review Assessment: Checking Correctness Of Derivations And Theory:**

I assessed the sensibility of the derivations and theory.

**Review Assessment: Checking Correctness Of Experiments:**

I assessed the sensibility of the experiments.

**Review Assessment: Thoroughness In Paper Reading:**

I read the paper at least twice and used my best judgement in assessing the paper.

---

### Author Response · Authors · 2019-11-14
**Main rebuttal (1)**

We would like to thank the reviewers for their review and start the discussion of our paper.
In the main rebuttal we would like to clarify novelty, the connection to other work, and why bandwidth-limited information transfer is a relevant problem.

# NOVELTY

In our paper, we assess the research question: How can we best model communication when the level of information transfer varies? This classic problem of information theory is highly relevant in many communication settings, as elaborated in the next section.

Specifically, in this work we attempt to understand what the best modelling choices are for this problem when using flexible function approximators to encode and decode messages. We make three fundamental observations about the model class that should be used.

For best message reconstruction, we observe:
  1. Bandwidth-limited communication should be modelled as a joint coding problem.
  2. Flexible approximate prior distributions should be provided for decoders to marginalize over possible message encodings.

When sampling from the communication model (e.g. when there is little transmitted information), we observe:
  3. Directly modelling the marginalization over missing information with an auxiliary latent variable decoder results in more reasonable (in distribution) decoded samples.

This assessment itself, and the resulting observations, are novel work to the best of our knowledge. Additionally, we propose a novel model for the bandwidth-limited channel (BWLC),as well as the novel concept of the auxiliary latent variable decoder.

# OTHER WORK

Our work is distinguished from other work in that it is the first to investigate the bandwidth-limited channel with the tools of machine learning.
Related work in this field focuses on the assessment of joint coding vs. separate coding (see last paragraph section 5) and channel coding (2nd paragraph section 5) with learned function approximators.


# APPLICATIONS

General
The BWLC model is applicable to many communication problems.
Typical examples involve communication over radio, telephone lines or WiFi. All three can be modelled as a BWLC with white noise. By extension, any radio signal, phone conversation, or internet traffic, such as video streaming, can be seen as a relevant application.

Reinforcement learning
It is also possible to integrate communication algorithms such as those proposed here into multi-agent reinforcement learning systems to emulate more realistic communication between agents.

Representation learning
Additionally, the bandwidth-limited channel orders the latent representation according to its importance for reconstruction. Thus, our approach could be useful for representation learning by allowing for straightforward dimensionality reduction. There may be a connection to other dimensionality reduction methods such as PCA that could be explored in follow up work.

---

### Decision · Program_Chairs · 2019-12-19

**Decision:**

Reject

**Comment:**

There was some support for this paper, but it was on the borderline and significant concerns were raised. It did not compare to the exiting related literature on communications, compression, and coding. There were significant issues with clarity.